# Low-Temperature Decomposition of Nitrous Oxide on Cs/Me$_x$Co$_{3-x}$O$_4$ (Me: Ni or Mg, x = 0–0.9) Oxides

## Yulia Ivanova and Lyubov Isupova *

Boreskov Institute of Catalysis, pr. Lavrentieva 5, 630090 Novosibirsk, Russia
* Correspondence: isupova@catalysis.ru

**Abstract:** Mixed oxides Me$_x$Co$_{3-x}$O$_4$ (Me: Ni or Mg, x = 0–0.9) with a spinel structure were synthesized by precipitation from Me, Co nitrate solutions using $(NH_4)_2CO_3$ as the precipitating agent with subsequent modification of the dry precipitate with cesium by the Pechini method and calcination. The samples were studied by XRD, TPR, and TPD methods. Their catalytic activity was studied in the low-temperature (150–350 °C) nitrous oxide decomposition process. It was shown that an increase in the degree of substitution of cobalt (x) leads to a significant decrease in the degree of crystallization of the oxides, an increase in the specific surface area, and the formation of surface weakly bound oxygen species. The highest activity was shown by the catalysts with a degree of substitution x = 0.1, especially by the nickel-substituted sample, which contained the maximum amount of weakly bound surface oxygen species. The difference in the influence of Mg and Ni on the Me$_x$Co$_{3-x}$O$_4$ properties is discussed.

**Keywords:** substituted cobalt spinel; nitrous oxide decomposition; loosely bound oxygen species

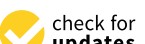



## 1. Introduction

Nitrous oxide ($N_2O$) is a greenhouse gas that has a high global warming factor, 310 times more than $CO_2$; thus, reducing $N_2O$ emissions is the current trend. Among various methods for removing nitrous oxide, the most interesting is the method of catalytic decomposition. One of the largest suppliers of $N_2O$ to the atmosphere is the production of nitric acid. The nitric acid production flow sheet includes several abatement methods/possibilities to reduce $N_2O$ emissions. Tertiary abatement measures of $N_2O$ removal after the absorber have great advantages because they do not affect the axial flow of nitric acid production, using a selective catalytic reduction (SCR) block with ammonia for de-NO$_x$ [1]. However, for implementation under reactor conditions, an SCR system requires a low-temperature catalyst. It is desirable that the de-$N_2O$ catalyst for the combined exhaust gas treatment system should work in the low-temperature range of 200–300 °C and be resistant to inhibitors like $O_2$ and $H_2O$. The range of catalytic systems for the low-temperature decomposition of $N_2O$ is wide [2,3] and includes platinum subgroup metals (Ru, Rh, Pd, Ir, and Pt), oxides of elements of the rare earth subgroup, and oxides of transition d-elements (in particular, Cu, Ni, Co, and Fe). Many catalytic systems containing noble metals and elements of the rare earth subgroup are deactivated in the presence of inhibitors and have a high cost, which makes it difficult to use them as the exhaust gas purification catalysts.

The catalytic decomposition of $N_2O$ (de-$N_2O$) implies that as a result, environmentally neutral molecules of $N_2$ and $O_2$ will be formed. The decomposition of $N_2O$ is an irreversible exothermic reaction. It proceeds by two mechanisms: the Langmuir–Hinshelwood (L–H) mechanism characterized by low activation energy of the stage of surface oxygen diffusion and recombination (stage 1) and the Eley–Rideal (E–R) mechanism characterized by high activation energy of surface diffusion, which requires large amounts of energy for oxygen diffusion (stage 3) [4–6]:

$$N_2O + S \rightarrow N_2\uparrow + S \ldots O_{surf} \tag{1}$$

$$2S \ldots O_{surf} \leftrightarrow 2S + O_2 \text{ (L-H)} \tag{2}$$

$$2S \ldots O_{surf} + N_2O \rightarrow 2S + O_2\uparrow + N_2\uparrow \text{ (E-R)} \tag{3}$$

where S is the active site of the surface, and S . . . O surf is the oxygen adsorbed on the active site of the surface. For various oxide systems, stages (2) and (3) were found to be relatively slow compared to stage (1). In other words, $O_2$ desorption can be considered the rate-determining stage of the catalytic decomposition of $N_2O$ regardless of the mechanism of $O_2$ formation. To date, the spinel-like structure of $Co_3O_4$ oxide, as well as the related binary substituted oxides, have proved to be the most promising candidates for de-$N_2O$ due to the redox ability of cobalt cations, a high concentration of oxygen vacancies, and a weak Co-O bond [7,8]. The substitution of Co cations in certain positions of the spinel oxide matrix has a significant effect on the binding energy of the active center with oxygen. The highest rates of activity in the decomposition reaction of $N_2O$ were demonstrated by magnesium- and nickel-substituted $Co_3O_4$ spinels. The incorporation of $Ni^{2+}$ or $Mg^{2+}$ into the $Co_3O_4$ structure promotes better desorption of $O_2$ and lowers the temperature of de-$N_2O$ [9–14]. Both cations can displace cobalt not only in the tetrahedral environment, but also to some extent in the octahedral one [15,16]. This leads to structural distortions with an increase in the bond lengths in octahedra and a decrease in tetrahedra. The effect is more pronounced for nickel-substituted $Co_3O_4$ spinels, and to a lesser extent for magnesium-substituted ones.

Surface modification of $Co_3O_4$ spinels with alkali metal cations makes them more active and resistant to inhibitors. The beneficial effect of different alkali promoters on the $Co_3O_4$ activity in nitrous oxide decomposition increases in the following order: Li $\ll$ Na < Rb $\cong$ K < Cs [17,18], which is consistent with a decrease in the electronegativity values from Li to Cs [19]. Most of the studies on the influence produced by the nature of the alkaline modifier and its precursors were carried out mainly for pure $Co_3O_4$ spinel. However, it cannot be ruled out that surface modification with akali cations of substituted $M_xCo_{3-x}O_4$ (Me = Mg, Ni) spinels may also change their catalytic activity and stability.

In this work, $Me_xCo_{3-x}O_4$ (Me: Ni or Mg, x = 0–0.9) oxides were prepared by the coprecipitation method. The samples were further modified with Cs cations according to the Pechini method. The influence of the catalyst composition on the catalytic activity and stability to inhibitors (oxygen, water vapor) in $N_2O$ decomposition was revealed.

## 2. Results and Discussion

### 2.1. XRD Study of Samples

Figure 1 shows X-ray diffraction patterns of the prepared 2%Cs/$Ni_xCo_{3-x}O_4$ (Cs/Ni-Co) and 1%Cs/$Mg_xCo_{3-x}O_4$ (Cs/Mg-Co) (x = 0–0.9) samples. The samples are well crystallized spinels (Fd-3m) [20] typical of $Co_3O_4$, and it is in good agreement with the available database (ICSD №69365). Table 1 lists the calculated parameters of crystal cells of the substituted cobalt spinels for the samples.

An increase in the Mg or Ni content leads to an increase in the lattice parameters and a decrease in the CSR (coherent scattering region) size, which is more pronounced in the case of Mg. On the X-ray diffraction patterns of Cs/Ni-Co (x = 0.5 and 0.9) samples, along with the peaks related to spinel, a reflection is additionally observed in the region of 2θ = 43.37, which refers to the NiO impurity phase. For Cs/Mg-Co samples, the formation of the magnesium oxide phase is not observed. The substitution with Mg or Ni leads also to an increase in $S_{BET}$ values of the samples (Table 1). In the case of substitution with Ni, the CSR size changes nonmonotonically with increasing the degree of substitution, as does $S_{BET}$. This can be attributed to the fact that a part of the nickel is not a part of the spinel but exists in the NiO impurity phase, while the formation of a continuous series of homogeneous solid solutions for the Cs/Mg-Co samples may be proposed [21].

According to [22], the formation of a normal spinel, where only $Co^{2+}$ cations are substituted, should not lead to an increase in the lattice parameter since the radii of the $Ni^{2+}$ (0.69 Å), $Mg^{2+}$ (0.72 Å) and $Co^{2+}$ (0.74 Å) cations are very close [23]. An increase in cell parameters with substitution degree indicates that both the cobalt and nickel

cations may occupy both the octahedral and tetrahedral positions. The inversion in both systems is in agreement with the literature data [22–25] Based on the presented data, $Mg_xCo_{3-x}O_4$ and $Ni_xCo_{3-x}O_4$ oxides are mostly the inverse spinels in which an increase in x leads to an increase in the length of octahedral bonds and a shortening of tetrahedral ones [14,15,24–27]. In [26], it was shown that as the spinel inversion increases, the lattice parameters can also increase.

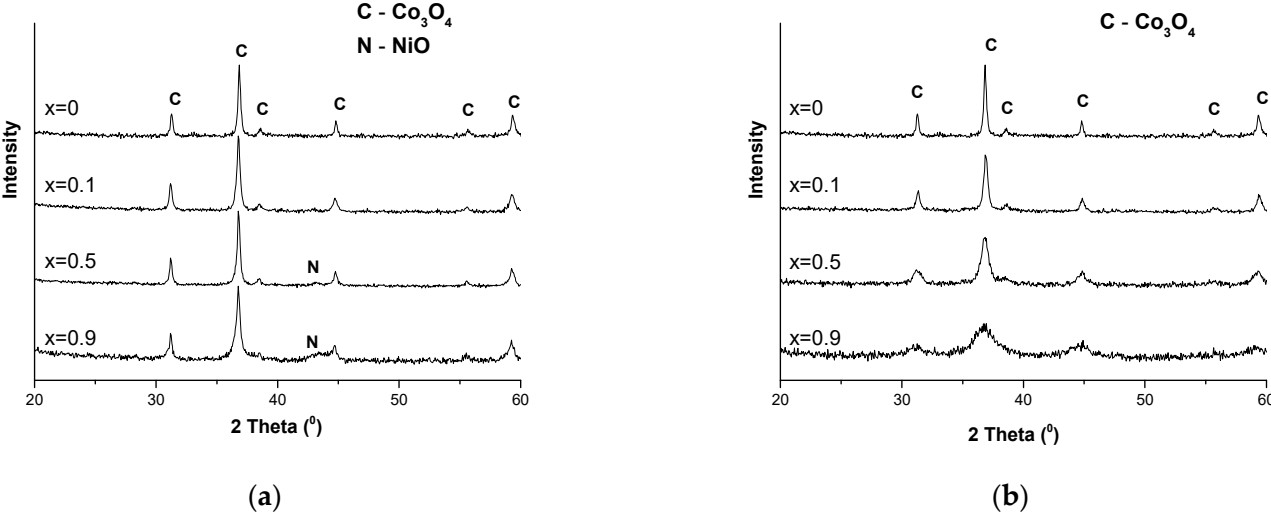

**Figure 1.** XRD patterns of samples: (**a**) Cs/Ni-Co; (**b**) Cs/Mg-Co.

Changes in the structural parameters and $S_{BET}$ in both substituted spinel systems Cs/Mg-Co and Cs/Ni-Co depend on the nature and amount of the modifying Ni or Mg cations. Thus, an increase in the lattice parameter of the prepared substituted spinels indicates their partial inversion.

**Table 1.** Physicochemical characteristics of the $Cs/Me_xCo_{3-x}O_4$ (Me: Ni or Mg, x = 0–0.9) samples and the half-transformation temperature ($T_{50}$).

| x | Phase Composition | Lattice Parameter, Å | CSR, Å | $S_{BET}$, $m^2/g$ | $T_{50}$, °C |
|---|---|---|---|---|---|
| 0 | Spinel | 8.083 | 380 | 12 | 275 |
| Cs/Ni-Co | | | | | |
| 0.1 | Spinel | 8.088 | 260 | 32 | 196 |
| 0.5 | Spinel, NiO | 8.094 | 310 | 23 | 224 |
| 0.9 | Spinel, NiO | 8.101 | 280 | 26 | 250 |
| Cs/Mg-Co | | | | | |
| 0.1 | Spinel | 8.084 | 210 | 36 | 210 |
| 0.5 | Spinel | 8.091 | 120 | 57 | 257 |
| 0.9 | Spinel | 8. 099 | 45 | 144 | 220 |

## 2.2. $H_2$-TPR and $O_2$-TPD Study of Samples

The temperature-programmed reduction ($H_2$-TPR) curves of the samples show two hydrogen uptake peaks (Figure 2a). The authors of [28,29] attribute the low-temperature peak to the reduction of $Co^{3+}$ to $Co^{2+}$, and the high-temperature peak to the reduction of $Co^{2+}$ to $Co^0$.

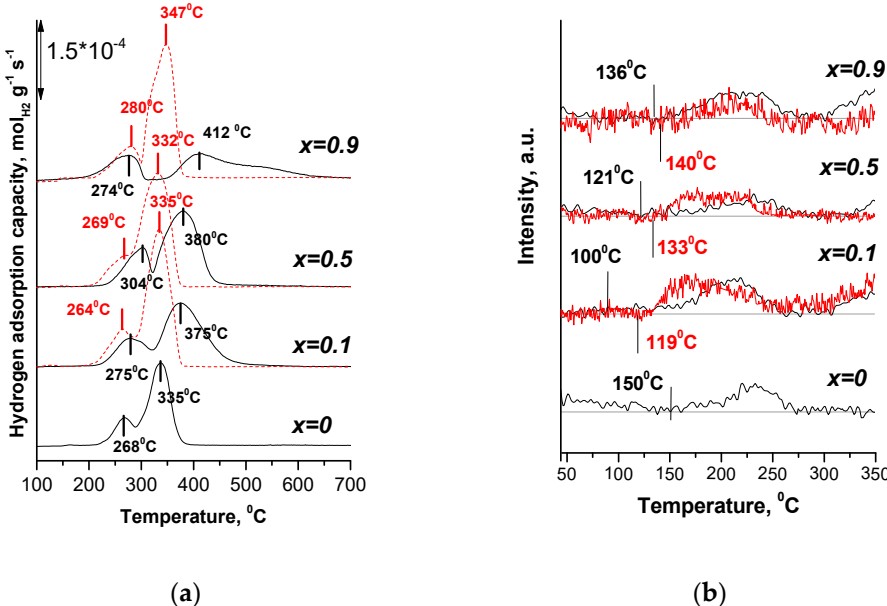

**Figure 2.** Profiles of the Cs/Mg-Co (black) and Cs/Ni-Co (red) samples obtained by (**a**) $H_2$-TPR and (**b**) $O_2$-TPD.

The substitution of cobalt cations with Ni/Mg cations lowers the initial reduction temperature ($T_0^{TPR}$) (Table 2). The temperatures of the reduction maxima corresponding to the gradual reduction of cobalt cations $Co^{3+} \rightarrow Co^{2+} \rightarrow Co^0$ were 268 and 335 °C, respectively, for the (x = 0) sample. These data are consistent with those reported in [28,29]. The reduction maxima of the low-temperature peaks for the substituted spinels shift insignificantly. In this case, the introduction of nickel leads to a decrease in the temperature of the maximum, and magnesium—to an increase. The shift of the maxima to the high-temperature region is most pronounced for the Cs/Mg-Co samples, but it virtually does not occur for the Cs/Ni-Co samples. According to the literature data for $Ni_xCo_{3-x}O_4$ [30,31], the temperature peaks at 221 and 284 °C correspond to the reduction of $Ni^{3+}$ to $Ni^{2+}$ and $Co^{3+}$ to $Co^{2+}$, respectively. As the temperature further increases, the reduction of $Ni^{2+}$ to $Ni^0$ and $Co^{2+}$ to $Co^0$ proceeds at the temperatures of 335 and 374 °C, respectively [30,31]. The authors of [21] state that depending on the CoO/MgO ratio, either (MgCo)O or $MgCo_2O_4$ solid solution can be formed. The reduction temperature of said compounds varies in the range of 500–1000 °C. Hence, it follows that the introduction of nickel has little effect on the reduction temperature of Cs/Ni-Co samples, while the introduction of magnesium increases the reduction temperature of Cs/Mg-Co samples and produces the particularly high temperature peaks. The increase in x leads to an increase in the octahedral bond lengths ($Me^{3+}$-O) and a shortening of the tetrahedral ($Me^{2+}$-O) ones [15]. This behaviour reflects the revealed cation distributions. The octahedra grow because of the increasing amount of larger divalent cations residing there. The smaller $Co^{3+}$ cations displaced by them enter the tetrahedral sites and the tetrahedral bond lengths decrease. Nickel in the lower degrees of substitution tends to occupy octahedral centers much more strongly than magnesium [15]. This may explain the greater increase in the lattice parameter and the decrease in the reduction temperature for the Cs/Ni-Co (x = 0.1) sample. The total amount of hydrogen consumed for the reduction of the $Cs/Co_3O_4$ sample is close to the theoretically calculated one, $16.6 \times 10^{-3}$ mol/g (Table 2). The total amount of hydrogen consumption decreases with an increase in the degree of substitution, the exception is the sample Cs/Ni-Co (x = 0.1). The value of hydrogen consumption in the first peak changes through a maximum in the degree of substitution (x = 0.1) for both series. Probably the increase in the absorption in the first peak for the (x = 0.1) samples is due to an increase in the content of $\Sigma(Co+Me)^{3+}$. In the second peak, the amount of absorbed hydrogen should be proportional to the amount of divalent cations, since the reduction of 1 mol of

$\Sigma(Co + Me)^{2+}$ accounts for 1 mol of $H_2$. The amount of absorbed hydrogen in the second peak for the Cs/Ni-Co (x = 0.1) sample exceeds the calculated amount of cations in the substituted samples $\Sigma(Co + Me)^{2+}$ = 12.5 × $10^{-3}$ mol/g for a normal spinel. Therefore, the reduction of $\Sigma(Co + Me)^{3+}$ for Cs/Ni-Co(x = 0.1) samples at low temperatures is not complete and continues at high temperatures, thus contributing to the absorption of $H_2$ in the second peak. This means that the actual amount of $\Sigma(Co+Me)^{3+}$ in the (x = 0.1) sample is greater than that calculated from the first peak. Thus, nickel is more prone to inversion. For the Cs/Mg-Co samples, the amount of absorbed $H_2$ in the second peak does not exceed the calculated amount of $\Sigma(Co+Me)^{2+}$; therefore, it reflects only the reduction of divalent cations. Thus, Cs/Ni-Co samples are more easily reduced by hydrogen and are more enriched in $\Sigma(Co+Me)^{3+}$ (taking into account that not all of them are restored in the first peak) than Cs/Mg-Co samples, especially in a low degree of substitution (x = 0.1).

According to the $O_2$-TPD data, the oxygen desorption occurs in two stages in a wide temperature range of 80–500 °C (Figure 2b). Considering the conditions of $N_2O$ decomposition in the SCR reactor, the low-temperature desorption peak is of the greatest interest. The initial temperature of $O_2$ desorption ($T_0^{TPD}$) for Cs/Co$_3$O$_4$ is 150 °C. The substitution of cobalt cations with Ni or Mg cations reduces the initial desorption temperature ($T_0^{TPD}$) in substituted Cs/Ni-Co and Cs/Mg-Co samples (Table 2). The presence of Mg in the samples reduces ($T_0^{TPD}$) more than Ni. The amount of desorbed oxygen varies through the maximum for both series at (x = 0.1), but is more pronounced for Cs/Ni-Co. At low temperatures, the main contribution to oxygen desorption is made by weak physical forces, for which an important role is played by the electronic interaction of the surface atoms of the catalyst, including Mg or Ni modifiers. The lower electronegativity of magnesium makes it possible to subsidize the electron density and reduce the energy of oxygen desorption. However, it is shown that a greater amount of desorbed oxygen in the Cs/Ni-Co samples is observed at relatively lower $S_{BET}$ values than in the Cs/Mg-Co samples. This leads to the imperfection of the oxygen sublattice and the formation of structural defects on which weakly-bound oxygen species are adsorbed. It is possible that this is the effect of a larger amount of applied cesium in the Cs/Ni-Co series. The $O_2$-TPD findings correlate with the $H_2$-TPR observations.

**Table 2.** $H_2$-TPR temperatures of the initial reduction and $H_2$ consumption. The amount of desorbed oxygen by $O_2$-TPD and the initial desorption temperatures.

| x | $T_0^{TPR}$, °C | $\Sigma H_2 \times 10^{-3}$, mol/g | $H_2 \times 10^{-3}$, mol/g $Co^{3+} \to Co^{2+}$ | $H_2 \times 10^{-3}$, mol/g $Co^{2+} \to Co^0$ | $T_0^{TPD}$, °C | $\Sigma O \times 10^{17}$, at/m$^2$ |
|---|---|---|---|---|---|---|
| 0 | 212 | 16.6 | 4.0 | 12.6 | 150 | 3.3 |
| | | | Cs/Ni-Co | | | |
| 0.1 | 188 | 17.1 | 4.2 | 12.9 | 119 | 4.2 |
| 0.5 | 200 | 15.7 | 3.3 | 12.4 | 133 | 3.9 |
| 0.9 | 207 | 15.5 | 3.5 | 12.0 | 140 | 1.2 |
| | | | Cs/Mg-Co | | | |
| 0.1 | 200 | 16.1 | 4.2 | 11.9 | 100 | 3.4 |
| 0.5 | 207 | 15.6 | 4.0 | 11.6 | 121 | 1.2 |
| 0.9 | 188 | 13.2 | 3.8 | 9.4 | 136 | 0.8 |

*2.3. Catalytic Activity*

2.3.1. Dependence of Activity on the Content of Cs

A study on the catalytic activity of Co spinel samples with a degree of substitution (x = 0.5) of Mg or Ni cations showed that when the cesium content varied from 0 to 3 wt%, samples containing 1–2 wt% Cs demonstrated high performance (Figure 3).

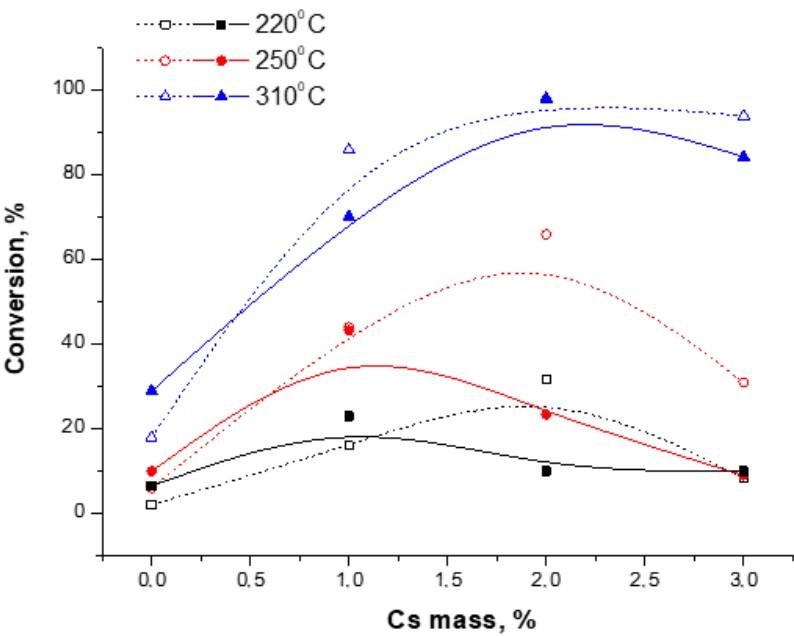

**Figure 3.** Influence of the mass content of Cs cations on $N_2O$ conversion in the temperature range 220–310 °C. Samples: $Ni_{0.5}Co_{2.5}O_4$ (dotted) and $Mg_{0.5}Co_{2.5}O_4$ (solid).

It was shown that at low temperatures of 220–250 °C, the optimal content is 1 wt% Cs for $Mg_{0.5}Co_{2.5}O_4$ and 2 wt% Cs for $Ni_{0.5}Co_{2.5}O_4$. The authors of [32] estimated the optimal content of potassium cations for the $Co_{2.6}Zn_{0.4}O_4$ catalyst in the range of 2–6 atoms of $K/nm^2$. According to [18,33], for the $Co_3O_4$ catalyst the optimal amount of Cs is 2–3 at. $Cs/nm^2$ (i.e., ~1 wt% Cs). In our case, the amount of the alkaline promoter does not correlate with the parameters ($S_{BET}$) of the samples: $Mg_{0.5}Co_{2.5}O_4$~1 at. $Cs/nm^2$ (1 wt% Cs) and for $Ni_{0.5}Co_{2.5}O_4$~4 at. $Cs/nm^2$ (2 wt% Cs). Thus, for a sample containing Ni, the mass content of Cs is two times higher despite the ~2.5 times smaller specific surface area compared to a sample containing Mg (Table 1). This can be explained by the lower Pauling electronegativity for Mg (0.65) in contrast to Ni (0.69), which further enhances the action of alkali metal cations. The low ionization energy of alkali metal cations allows them to easily form a bond with oxygen adsorbed on the surface of the catalyst, promoting the recombination of active centers of $N_2O$ decomposition [34]. Magnesium, unlike nickel, is the main donor of electron density. The properties of alkaline earth metal cations are close to alkaline, which causes a decrease in the concentration of Cs as a promoter for $Mg_{0.5}Co_{2.5}O_4$.

Based on the data obtained, a series of Cs/Mg-Co samples was modified with 1 wt% Cs, and a series of Cs/Ni-Co samples—with 2 wt% Cs.

### 2.3.2. The Decomposition of $N_2O$

A study on the catalytic activity of Cs/Mg-Co and Cs/Ni-Co samples with cobalt substitution degrees (x = 0–0.9) in a model mixture containing 0.15% $N_2O$ in helium showed that high conversion values were demonstrated by samples with the (x = 0.1) substitution degree (Figure 4).

The half-transformation temperature ($T_{50}$) for samples with the degree of substitution (x = 0.1) is minimal and amounts to 196 and 210 °C for Cs/Ni-Co and Cs/Mg-Co, respectively (Table 1). For unsubstituted cobalt spinel $Co_3O_4$, this value is maximal and amounts to 275 °C. High specific activity (the activity normalized per $m^2$, the reaction rate) was demonstrated by Cs/Ni-Co (x = 0.1 and 0.5) and Cs/Mg-Co (x = 0.1) samples (Figure 4b). The Cs/Mg-Co sample with (x = 0.1) is inferior to the Cs/Ni-Co samples (x = 0.1 and 0.5) in the reaction rate and in the amount of desorbed oxygen referred to the surface area. The observed correlation indicates a significant effect of weakly bound surface oxygen in the

catalysts on the reaction rate, which is consistent with the data on the rate-limiting stage of this reaction at low temperatures [4,5].

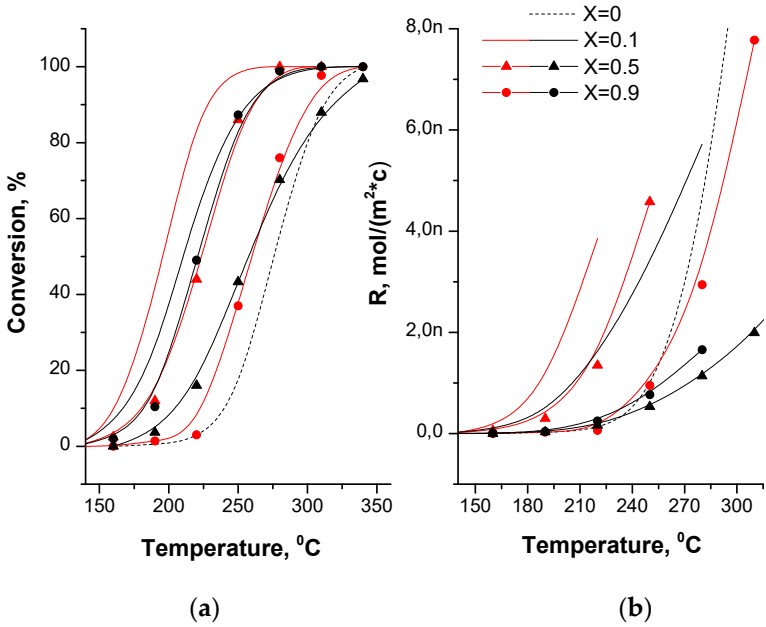

**Figure 4.** Temperature dependence of (**a**) conversion and (**b**) reaction rate of $N_2O$ decomposition for Cs/Mg-Co (black) and Cs/Ni-Co (red) samples.

### 2.3.3. The Decomposition of $N_2O$ in the Presence of Inhibitors

Under the actual conditions of the SCR reactor, the tail gases from the production of nitric acid, in addition to nitrous oxide, contain oxygen and water vapor [5]. The effect of inhibitors ($H_2O$, $O_2$) was tested on samples with high activity in the decomposition of pure nitrous oxide Cs/Ni-Co and Cs/Mg-Co with the degree of substitution (x = 0.1). The data obtained indicate a decrease in the catalyst activity in the presence of inhibitors in the reaction mixture (Figure 5). Samples Cs/Ni-Co and Cs/Mg-Co (x = 0.1) lose their activity during the first two hours in the reaction mixture of 1500 ppm $N_2O$ in He (without inhibitors), nitrous oxide conversion for Cs/Ni-Co decreases by ~15% and for Cs/Mg-Co by ~7%. In the presence of inhibitors ($H_2O$, $O_2$), the initial activity of the samples is lower than in their absence, but there is a tendency to lose activity in the first hours. The loss of activity was ~25% for Cs/Ni-Co and ~10% for Cs/Mg-Co. In this case, the unmodified Cs/$Co_3O_4$ catalyst loses ~40% of its activity in the presence of inhibitors. Long-term tests revealed a higher stability of the Cs/Mg-Co sample as compared to Cs/Ni-Co. A similar effect of stronger inhibition on the $Ni_{0.74}Co_{0.26}Co_2O_4$ catalyst than on $Mg_{0.54}Co_{0.46}Co_2O_4$ was observed in [9]. The authors of [9,35–37] showed a stronger inhibition of substituted Co spinels by water rather than by oxygen.

A comparison of the $N_2O$ conversion and the specific activity of Cs/Ni-Co and Cs/Mg-Co samples shows that their activity increases not only due to an increase in $S_{BET}$, but mainly due to an increase in the number of weakly bound oxygen centers of the samples. The Cs/Ni-Co (x = 0.1) sample was characterized by the maximum values of oxygen desorption according to $O_2$-TPD and $H_2$-TPR data and had high conversion and specific activity normalized to the specific surface area value. The high activity of the Cs/Ni-Co (x = 0.1) sample is due to a large number of weakly bound oxygen centers. The same active sites interact with inhibitors, which leads to rapid deactivation of the Cs/Ni-Co (x = 0.1) sample compared to Cs/Mg-Co (x = 0.1). The deactivation of both samples in the presence of inhibitors or without them proceeds in the first ~2 h.

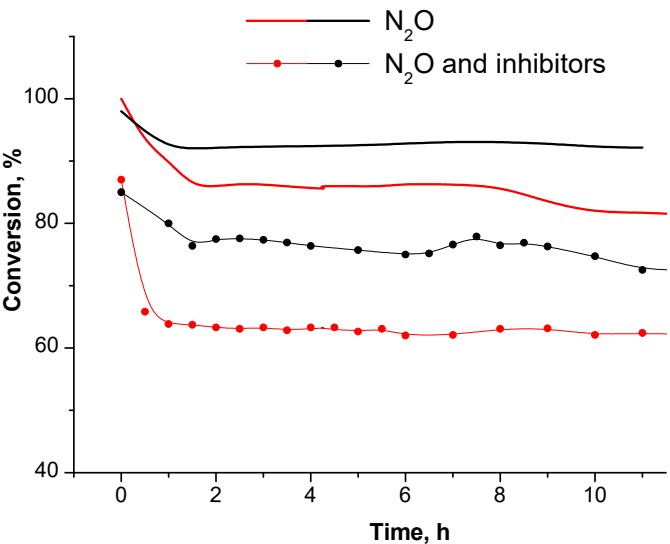

**Figure 5.** The $N_2O$ conversion during testing of the Cs/Mg-Co (black) and Cs/Ni-Co (red) (x = 0.1) samples at 300 °C.

## 3. Experimental

### 3.1. Samples Preparation

Cs/$Me_xCo_{3-x}O_4$ (x = 0–0.9, Me = Mg, Ni) samples were synthesized by coprecipitation. Short designations for the samples are Cs/Mg-Co and Cs/Ni-Co, respectively. For the synthesis, $Ni(NO_3)_2 \cdot 6H_2O$ or $Mg(NO_3)_2 \cdot 6H_2O$ and $Co(NO_3)_2 \cdot 6H_2O$ aqueous solutions were used in the stoichiometric ratio of cations required to obtain a mixed spinel of a given composition. Precipitation was implemented at room temperature and a pH of 8–8.5 using $(NH_4)_2CO_3$ as the precipitating agent. The resulting precipitates were filtered off, washed to a pH of 7 and dried at 120 °C for 10 h. The dried precipitates were modified with cesium by impregnation from a solution containing $CsNO_3$, ethylene glycol, and citric acid (the Pechini method). The Cs/Mg-Co samples were modified with 1 wt% Cs; the Cs/Ni-Co samples and the $Cs/Co_3O_4$ sample were modified with 2 wt% Cs. All the samples were calcined for 2 h at a temperature of 450 °C.

### 3.2. Catalysts Characterization

The phase composition of the samples was determined by X-ray diffraction (XRD) analysis on a Bruker D8 diffractometer using CuK α radiation (λ = 1.5418 Å). The samples were scanned point-by-point with 0.05° increments in a 2θ range of 20–60°. The structural parameters were calculated using the FullProff software. The specific surface area ($S_{bet}$, $m^2$/g) was measured by argon sorption at −196 °C, followed by thermal desorption, at four points of the sorption equilibrium with a SORBI m 4.1 device (ZAO META) using the Sorbi m Version 4.2 software. Helium was used as a carrier gas. The values of $S_{bet}$ were calculated by the Brunauer–Emmett-Teller method (BET). The samples with the grain size of 0.25–0.50 mm were studied by a temperature programmed reduction with hydrogen ($H_2$-TPR) in a flow unit equipped with a thermal conductivity detector for a catalyst fraction of 0.25–0.50 mm. A weighed sample (10 mg) was subjected to preliminary holding in argon at 200 °C for 30 min and subsequent cooling in argon to room temperature, and then reduced in a mixture (10% $H_2$ in argon) at a flow rate of 40 mL/min. The samples were heated to 900 °C at a rate of 10 deg/min. Studies of the samples by temperature programmed desorption of oxygen ($O_2$-TPD) were conducted in a flow unit. The mixture at the outlet of the reactor was analyzed using a QMS 100 mass spectrometer (Stenford Research Systems, SRS). The samples were pretreated in a mixture of 20% $O_2$ in He at 450 °C for 60 min and then cooled to room temperature. The weighed portion of the sample was 200 mg; the feed flow rate of He was 3.6 L/h. The samples were heated to 450 °C at a rate of 10 °C/min.

### 3.3. Activity Tests

The catalytic activity in de-$N_2O$ reaction was studied in a flow tube reactor (5 mm inner diameter) at 130–430 °C, ambient pressure, and a space velocity (GHSV) of 9000 h$^{-1}$. Standard tests were carried out using the reaction mixture of 1500 ppm $N_2O$ in He. Stability experiments were carried out in a gas mixture of 3% $O_2$ or 2.5–3% $H_2O$ at a temperature of 250 °C. Reactant and product concentrations were analyzed by a Fourier transform IR spectrometer FT-801. The conversion of $N_2O$ was calculated by the following equation:

$$X_{N_2O} = \frac{C_{N_2O}^{decomp}}{C_{N_2O}^{feed}} \times 100\%,$$

where $C_{N_2O}^{decomp}$ is the decomposed gas concentration and $C_{N_2O}^{feed}$ is the feed gas concentration. The rate was determined under the assumption of a first-order reaction by the following equation:

$$R_{N_2O} = \frac{UN_A}{mS_{sp}} \, ln\frac{1}{(1-x)} \; [\text{mol } m^{-2} s^{-1}],$$

where $U$ is the flow rate of the reaction mix [mol/s]; $N_A$ is the Avogadro constant; $m$ is the weight [g]; $S_{sp}$ is the specific surface area [m$^2$/g], and $x$ is the fraction of converted $N_2O$.

The temperature at which the $N_2O$ conversion was 50% (denoted as $T_{50}$—the half-life temperature of nitrous oxide) was a measure of the catalyst activity.

### 4. Conclusions

A series of 1%Cs/Mg$_x$Co$_{3-x}$O$_4$ and 2%Cs/Ni$_x$Co$_{3-x}$O$_4$ oxide catalysts with the degree of substitution (x = 0–0.9) prepared by precipitation route and promoted with Cs had the spinel structure typical of Co$_3$O$_4$. The change in the structural parameters and S$_{bet}$ in both series depends on the nature and amount of modifying Ni or Mg cations. The observed increase in the lattice parameters of substituted spinels in both series indicates their partial inversion.

Evaluation of the influence of the promoter on the activity of substituted spinels for (x = 0.5) samples showed that the optimal amount of the promoter depends on the nature of the substituting cation. The optimal amount of alkaline promoter for the samples containing Mg was determined as 1 wt% Cs, while for the samples containing Ni, as 2 wt% Cs.

2%Cs/Ni$_{0.1}$Co$_{2.9}$O$_4$ samples are more easily reduced with hydrogen (H$_2$-TPR) than 1%Cs/Mg$_{0.1}$Co$_{2.9}$O$_4$ samples and contain a large quantity of weakly bound oxygen species according to O$_2$-TPD and H$_2$-TPR. 1%Cs/Mg$_{0.1}$Co$_{2.9}$O$_4$ and 2%Cs/Ni$_{0.1}$Co$_{2.9}$O$_4$ samples with the degree of substitution (x = 0.1) showed the highest conversion degrees in the series in $N_2O$ decomposition, which correlates with H$_2$-TPR and H$_2$-TPD data. Therewith, the 2%Cs/Ni$_{0.1}$Co$_{2.9}$O$_4$ samples (x = 0.1 and 0.5) had a higher specific activity as compared to the 1%Cs/Mg$_{0.1}$Co$_{2.9}$O$_4$ samples. The activity increases mainly due to the large number of weakly bound oxygen species.

Under the conditions of a SCR reactor, the 1%Cs/Mg$_{0.1}$Co$_{2.9}$O$_4$ sample is of the greatest interest for application as the second layer of the catalyst for the decomposition of nitrous oxide, taking into account the resistance of the samples to the reaction medium.

**Author Contributions:** Y.I.—methodology, conceptualization, validation, and investigation; L.I.—supervision, methodology. All authors have read and agreed to the published version of the manuscript.

**Funding:** This work was supported by the budget project AAAA-A21-121011490008-3 for the Boreskov Institute of Catalysis.

**Data Availability Statement:** Informed consent was obtained from all subjects involved in the study.

**Conflicts of Interest:** The authors declare no conflict of interest.

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
