# Peer review of "Low-Temperature Decomposition of Nitrous Oxide on Cs/MexCo3−xO4 (Me: Ni or Mg, x = 0–0.9) Oxides"

_catalysts, doi:10.3390/catal13010137_

Round 1

Reviewer 1 Report

I have reviewed the submitted paper to Catalysts/MDPI entitled “Low-Temperature Decomposition of Nitrous Oxide on Cs/MexCo3-xO4 (Me: Ni or Mg, x = 0–0.9) Oxides” Isupova et al.

Nitrous oxide (N2O) contributes to the depletion of the ozone layer and contributes to the greenhouse effect due to its high ability to absorb and backscatter thermal radiation to the earth. Thus, it poses a huge global warming potential. A promising way to decrease N2O emissions from nitric oxide production is the catalytic decomposition of nitrous oxide into nitrogen and oxygen in a tertiary alignment. Therefore, in the literature, many types of mixed oxides and zeolites were examined and reported. Most of them suffer from deactivation in the presence of O2, NO and H2O in the reactant stream or they require high temperatures for high conversions. Cobalt-based spinels bear a high potential as feasible catalysts for N2O decomposition.

In the submitted work, the authors have coined a new catalyst using various metals/alkali metal cations as dopants and their influence on the catalytic activity of the titled materials were examined. The results showed that the alkali cation substitution also change the catalytic activity and stability.

The manuscript is written well, however, the areas that require clarification are given below.

My specific points are:

·         How the current work differs to that of the reported nickel-cobalt mixed oxides in the literature (doi.org/10.3390/catal12111405) for N2O decomposition part from dopant variation.

·         The reactions such as dissociative adsorption and desorption during N2O decomposition must be provided.

·         Is that due to the dopants, the number of active sites in CO3O4 has been increased that influences the catalytic activity?

·         Do the Co atoms are distributed in octehdral (or) tetrahedral sites?

·         The reduction of Co3+ towards Co2+ and the respective XRD patterns can be referred to the database reported in the literature (such as doi.org/10.3390/nano7110356; 10.1039/D0NJ01486A)

·         With regard to a commercial application, can this material (dopant added) be an interesting candidate?

·         Removing NOx prior to the deN2O stage makes the addition of reducing agents (NH3) not required and suitable for marketing.

·         The synergistic effect between dopants (Cs/Ni) and Co atoms in the catalyst must be detailed. 

Author Response

The List of Changes and Comments to Referees’ Remarks

Manuscript ID: catalysts-2124150

Title: Low-Temperature Decomposition of Nitrous Oxide on Cs/MexCo3-xO4 (Me: Ni or Mg, x = 0–0.9) Oxides

The authors are very grateful to the reviewers for their in-depth and careful review. We have revised our present paper in the light of their useful suggestions and comments. We hope our revision has improved the paper to a level of their satisfaction. The answers to their specific comments/suggestions are as follows

Reviewer #1:

I have reviewed the submitted paper to Catalysts/MDPI entitled “Low-Temperature Decomposition of Nitrous Oxide on Cs/MexCo3-xO4 (Me: Ni or Mg, x = 0–0.9) Oxides” Isupova et al.

Nitrous oxide (N2O) contributes to the depletion of the ozone layer and contributes to the greenhouse effect due to its high ability to absorb and backscatter thermal radiation to the earth. Thus, it poses a huge global warming potential. A promising way to decrease N2O emissions from nitric oxide production is the catalytic decomposition of nitrous oxide into nitrogen and oxygen in a tertiary alignment. Therefore, in the literature, many types of mixed oxides and zeolites were examined and reported. Most of them suffer from deactivation in the presence of O2, NO and H2O in the reactant stream or they require high temperatures for high conversions. Cobalt-based spinels bear a high potential as feasible catalysts for N2O decomposition.

In the submitted work, the authors have coined a new catalyst using various metals/alkali metal cations as dopants and their influence on the catalytic activity of the titled materials were examined. The results showed that the alkali cation substitution also change the catalytic activity and stability.

The manuscript is written well, however, the areas that require clarification are given below.

My specific points are:

  • How the current work differs to that of the reported nickel-cobalt mixed oxides in the literature (doi.org/10.3390/catal12111405) for N2O decomposition part from dopant variation.

Response. In the work [doi.org/10.3390/catal12111405], the samples of the only composition Ni/Co~0.32 but with different structural properties, which were obtained as a result of different preparation methods, were studied. We studied the samples of different composition obtained by the same method. The fundamental difference is that our samples were in addition modified with Cs+.

The conditions for testing the samples differed significantly as well: in the proposed work, the gas mixture contained 5 or 20% N2O in the presence or absence of oxygen (1:1) (WHSV=15,000  h-1). The authors indicate that they are interested in catalytic systems capable of successfully treat feeding streams containing high concentrations of N2O, such as in the case of adipic acid production plants. These conditions strongly differ from conditions of SCR reactor that were used in our experiments due to the main goal of our paper.  

  • The reactions such as dissociative adsorption and desorption during N2O decomposition must be provided.

Response. Agree.

Action taken. The manuscript was corrected. Equations describing the stages of the nitrous decomposition process have been added. Phrases describing the stages of the mechanism are added (”1. Introduction”).

The catalytic decomposition of N2O (de-N2O) implies that as a result, environmentally neutral molecules of N2 and O2 will be formed. The decomposition of N2O is an irreversible exothermic reaction. It proceeds by two mechanisms: Langmuir–Hinshelwood (L–H) mechanism characterized by low activation energy of the stage of surface oxygen diffusion and recombination (stage 1) and Eley–Rideal (E–R) mechanism characterized by high activation energy of surface diffusion, which requires large amounts of energy for oxygen diffusion (stage 3) [4-6]:

N2O + S → N2↑ + S…Osurf                              (1),

2S…Osurf « 2S + O2(L-H)                              (2),

2S … O surf   +  N2O → 2 S + O2 + N2  (E-R)               (3),

where S is the active site of the surface, and S…O surf  is the oxygen adsorbed on the active site of the surface. For various oxide systems, stages (2) and (3) were found to be relatively slow compared to stage (1). In other words, O2  desorption can be considered the rate-determining stage of the catalytic decomposition of N2O regardless of the mechanism of O2  formation.”

[4-6] https://doi.org/10.1016/j.catcom.2017.12, https://doi.org/10.1006/jcat.1997.1581, https://doi.org/10.1016/S0926-860X(97)00357-8

  • Is that due to the dopants, the number of active sites in Co3O4 has been increased that influences the catalytic activity?

Response. Yes. The inclusion of Ni2+ or Mg2+ in the Co3O4 structure leads to the formation of structural defects (additional active centers) capable of adsorbing of oxygen. This results in higher deN2O activity.

  • Do the Co atoms are distributed in octahedral (or) tetrahedral sites?

Response. According to literature data [Liu, X.;Prewitt, C.T./ High-temperature X-ray diffraction study of Co3O4: transition from normal to disordered spinel.// Phys Chem Minerals (1990) 17:168-172] at room temperature, cubic Co3O4 is a normal spinel with the Co2+ ions occupying the tetrahedral sites, and Co3+ ions the octahedral sites. Substitution of Me2+ cations for cobalt in spinel leads to inversion, where а partial substitution of Co3+ for Co2+ in tetragonal positions is possible then the charge balance is compensated by the corresponding substitution of Co2+ for Co3+  in octahedral sites. The inversion in cations distributions result in imperfection of the oxygen sublattice that influence the catalytic activity.

  • The reduction of Co3+ towards Co2+ and the respective XRD patterns can be referred to the

database reported in the literature (such as doi.org/10.3390/nano7110356; 10.1039/D0NJ01486A)

Response. Thanks for the interesting information.

We did not obtain an X-ray diffraction pattern of the oxide reduced in the H2-TPR process to CoO, since the main task was to evaluate the reactivity of oxygen. The rationale for the reduction in the first  peak to CoO is the amount of absorbed hydrogen that is in accordance with the equation Co3O4CoO.

 Оur samples have the cubic structure of the Co3O4  space group (Fd-3m), which are in good agreement with the available database (ICSD № 69365)

Action taken. We referred to the publication (doi.org/10.3390/nano7110356) in the (“2.1. XRD study of samples”) part and indicated available database (ICSD № 69365) for structure of the Co3O4.

  • With regard to a commercial application, can this material (dopant added) be an interesting candidate?

Response. Yes, we are working on this issue of a commercial application. There is our work with a mathematical model describing the possibilities of the realization of integrated low-temperature catalytic purification of gases formed in the production of nitric acid (tail gases) from nitrogen oxides (NO, NO2) and nitrogen (I) oxide (N2O) in one reactor [doi.org/10.15372/CSD2020221].

  • Removing NOx prior to the deN2O stage makes the addition of reducing agents (NH3) not required and suitable for marketing.

Response.  Yes, that's why we settled on an N2O decomposition catalyst to remove it.  First of all, purification of exhaust gases in the production of nitric acid implies the removal of highly toxic nitrogen oxides NOх (NO and NO2) because their average daily maximum permissible concentrations (MPC) in the atmosphere are  only 0.04 and 0.06 mg/m3 , respectively. There are several ways to remove NOx from the tail gases:

First of all, purification of exhaust gases in the production of nitric acid implies the removal of highly toxic nitrogen oxides NOх (NO and NO2) because their average daily maximum permissible concentrations (MPC) in the atmosphere are  only 0.04 and 0.06 mg/m3 , respectively. There are several ways to remove NOx from the tail gases:

Hightemperature reduction with СН4 at a temperature around 700 °С.  Not only NOх  but also N2O is reduced in this process;

Non-selective catalytic reduction (in the absence of oxygen), possible reducing gases may be purge gas of ammonia cycle (mainly Н2), hydrocarbons  (natural  gas,  propane,  butane).  Not only NOх  but also N2O is reduced in this process;

Selective  non-catalytic reduction proceeds at a temperature above 800 °C, under these conditions NOх  is reduced by ammonia without any catalysts;  

Selective catalytic reduction (SCR) of NOх with ammonia proceeds in the presence of oxygen and takes place at a temperature of 200–450 °C. The use of a catalyst allows NOх  reduction at substantially lower temperature. This low-temperature NOx removal process is implemented in most UKL technologies and requires additional N2O removal. We suggest N2O removal for this case by placing the N2O decomposition catalyst bed after the SCR catalyst bed. 

Reviewer 2 Report

This manuscript used coprecipitation and sol-gel methods to achieve Cs/MexCo3-xO4 spinel oxides. The achieved samples were characterized by XRD, TPR, TPD and N2 absorption/desorption techniques. Their structures and catalytic tests were further analyzed in details, which may provide some guidance for design new catalysts for N2O decomposition application. However, there are several points need to address clearly before being accepted.

1.The author claimed SEM techniques for achieved samples in abstract, but failed to show them in the main text. Besides, I would prefer to use N2 adsorption/desorption technique rather than BET method. The N2 absorption/desorption curves were missing in the manuscript.

2. How about their HRTEM and corresponding EDX data? It would help a lot to see the distribution of Cs component and other metals.

3.The XRD pattern before dry-precipitate with cesium is missing. Is there any difference before and after Cs precipitation?

4.It’s confusing to use different abbreviation names/types with the same samples. For example, Cs/MexCo3-xO4 was used in the title and abstract, but in results and discussion part, Cs/Ni-Co and Cs/Mg-Co was used instead. Please consolidate them.

5.For TPR, the author claimed that 284C corresponds to the reduction of Co3+ to Co2+,but why was it 268C when x=0? What’s the cause of the difference?

6.What’s the production of decomposition of N2O? Please state that clearly in the manuscript.

7.The experiment parts are not stated clearly. For example, How much of chemical precursors were used for synthesizing MexCo3-xO4? What’s the amount of Cs precursor used to produce different mass loading of Cs on spinel supports? Also, why did the author use CeNO3 to produce Cs/MexCo3-xO4? It seems so abnormal.

Author Response

The List of Changes and Comments to Referees’ Remarks

Manuscript ID: catalysts-2124150

Title: Low-Temperature Decomposition of Nitrous Oxide on Cs/MexCo3-xO4 (Me: Ni or Mg, x = 0–0.9) Oxides

The authors are very grateful to the reviewers for their in-depth and careful review. We have revised our present paper in the light of their useful suggestions and comments. We hope our revision has improved the paper to a level of their satisfaction. The answers to their specific comments/suggestions are as follows

Reviewer #2:

This manuscript used coprecipitation and sol-gel methods to achieve Cs/MexCo3-xO4 spinel oxides. The achieved samples were characterized by XRD, TPR, TPD and N2 absorption/desorption techniques. Their structures and catalytic tests were further analyzed in details, which may provide some guidance for design new catalysts for N2O decomposition application. However, there are several points need to address clearly before being accepted.

1.The author claimed SEM techniques for achieved samples in abstract, but failed to show them in the main text. Besides, I would prefer to use N2 adsorption/desorption technique rather than BET method. The N2 absorption/desorption curves were missing in the manuscript.

Response. Agree. SEM  is typo. Nitrogen, argon, or krypton is often chosen as an adsorbent to determine the surface area of porous materials.

Action taken. The manuscript was corrected as following: The abbreviations “SEM and BET “ were removed from the abstract.  The phrase regarding “SEM“  was removed from the part “3.2. Catalysts characterization”.  

The methodology was added to the (“3.2. Catalysts characterization”) part.

“The specific surface area (Sbet, m2g–1) was measured by argon sorption at -196 °C, followed by thermal desorption, at four points of the sorption equilibrium with a SORBI-M 4.1 device (ZAO META) using the Sorbi-M Version 4.2 software. Helium was used as a carrier gas. The values of Sbet were calculated by the Brunauer– Emmett- Teller method (BET).

  1. How about their HRTEM and corresponding EDX data? It would help a lot to see the distribution of Cs component and other metals.

Response. Agree. This is, of course, very valuable information, but unfortunately we cannot get it quickly due to administrative problems.

3.The XRD pattern before dry-precipitate with cesium is missing. Is there any difference before and after Cs precipitation?

Response.  Experiments of this kind were not  carried out on dried precipitates (carbonates), but the XRD patterns were obtained for samples after calcination (Fig.1, Tablе). Precipitation of Cs leads to an decrease in the spinel lattice parameters, Sbet  and opposite to an increase in the X-ray particle size.

Figure 1. XRD patterns of samples: Ni0.1Co 2.9O4 and 2%Cs/Ni0.1Co 2.9O4.

Table. Physicochemical characteristics of the Ni0.1Co 2.9O4 and 2%Cs/Ni0.1Co 2.9O4 samples

Samples

Sbet

m2/g

Phase composition

Lattice parameter,

a=b=c, Å

CSR, Ǻ

Ni0.1Co 2.9O4

56

Co3O4

8.110

150

2%Cs/Ni0.1Co 2.9O4

32

Co3O4

8.088

260

4.It’s confusing to use different abbreviation names/types with the same samples. For example, Cs/MexCo3-x O4 was used in the title and abstract, but in results and discussion part, Cs/Ni-Co and Cs/Mg-Co was used instead. Please consolidate them.

Response. Agree.

Action taken. The abbreviated names (Cs/Ni-Co and Cs/Mg-Co) are indicated  in the text of paragraph “2. Results and Discussion”. Corrections were made in the part “4. Conclusions”, namely, Cs/Ni-Co and Cs/Mg-Co are replaced by Cs/MexCo3-xO4.

5.For TPR, the author claimed that 284C corresponds to the reduction of Co3+ to Co2+,but why was it 268C when x=0? What’s the cause of the difference?

Response. The reference sample (with x=0 ) was modified with 2 wt% Cs (it is indicated in the “3.1. Samples preparation” part). The authors of [doi.org/10.1016/j.apcatb.2015.04] described mesoporous samples not modified with alkali metals. The decrease in the reduction temperature for our sample with х=0 but modifed with Cs  is due to the influence of cesium.

6.What’s the production of decomposition of N2O? Please state that clearly in the manuscript.

Response. Agree.

Action taken. The manuscript was corrected. Equations describing the stages of the nitrous decomposition process have been added. Phrases describing the stages of the mechanism are added (”1. Introduction”).

The catalytic decomposition of N2O (de-N2O) implies that as a result, environmentally neutral molecules of N2 and O2 will be formed. The decomposition of N2O is an irreversible exothermic reaction. It proceeds by two mechanisms: Langmuir–Hinshelwood (L–H) mechanism characterized by low activation energy of the stage of surface oxygen diffusion and recombination (stage 1) and Eley–Rideal (E–R) mechanism characterized by high activation energy of surface diffusion, which requires large amounts of energy for oxygen diffusion (stage 3) [4-6]:

N2O + S → N2↑ + S…Osurf                              (1),

2S…Osurf « 2S + O2(L-H)                              (2),

2S … O surf   +  N2O → 2 S + O2 + N2  (E-R)               (3),

where S is the active site of the surface, and S…O surf  is the oxygen adsorbed on the active site of the surface. For various oxide systems, stages (2) and (3) were found to be relatively slow compared to stage (1). In other words, O2  desorption can be considered the rate-determining stage of the catalytic decomposition of N2O regardless of the mechanism of O2  formation.”

[4-6] https://doi.org/10.1016/j.catcom.2017.12, https://doi.org/10.1006/jcat.1997.1581, https://doi.org/10.1016/S0926-860X(97)00357-8

  1. The experiment parts are not stated clearly. For example, How much of chemical precursors were used for synthesizing MexCo3-xO4? What’s the amount of Cs precursor used to produce different mass loading of Cs on spinel supports? Also, why did the author use CeNO3 to produce Cs/MexCo3-xO4? It seems so abnormal.

 Response. The raw material for precipitation were taken to obtain 20 g of MexCo3-xO4 The amount of CsNO3 precursor used for modification of dry precipitate was calculated as the mass percentage of Cs (metal) to the mass of the oxide that forms after calcination of precipitate.

Action taken. It was a typo (CeNO3), it was corrected (CsNO3)

Round 2

Reviewer 1 Report

The revised manuscript is OK to publish.

Reviewer 2 Report

The revised version made sufficient correction based on previous comments and meets the publicationcriterion on this journal though there are still lack of some morphology checking techniques such as SEM, TEM..... I totally understand that it might due to the insuficient instrument there. So I prefer to accept this manuscript for publication.